# Predictive Evaluation on Cytological Sample of Metastatic Melanoma: The Role of BRAF Immunocytochemistry in the Molecular Era

**DOI:** 10.3390/diagnostics11061110

**Published:** 2021-06-18

**Authors:** Andrea Ronchi, Marco Montella, Federica Zito Marino, Michele Caraglia, Anna Grimaldi, Giuseppe Argenziano, Elvira Moscarella, Gabriella Brancaccio, Teresa Troiani, Stefania Napolitano, Renato Franco, Immacolata Cozzolino

**Affiliations:** 1Pathology Unit, Department of Mental and Physical Health and Preventive Medicine, Università degli Studi della Campania “Luigi Vanvitelli”, 80138 Naples, Italy; marco.montella@unicampania.it (M.M.); federica.zitomarino@unicampania.it (F.Z.M.); coimma73@gmail.com (I.C.); 2Department of Precision Medicine, University of Campania “Luigi Vanvitelli”, 80138 Naples, Italy; michele.caraglia@unicampania.it (M.C.); anna.grimaldi@unicampania.it (A.G.); 3Dermatology Unit, Department of Mental and Physical Health and Preventive Medicine, Università degli Studi della Campania “Luigi Vanvitelli”, 80138 Naples, Italy; giuseppe.argenziano@unicampania.it (G.A.); elvira.moscarella@unicampania.it (E.M.); gabri.brancaccio@gmail.com (G.B.); 4Oncology Unit, Department of Precision Medicine, Università della Campania “Luigi Vanvitelli”, 80131 Naples, Italy; teresa.troiani@unicampania.it (T.T.); stefania.napolitano@unicampania.it (S.N.)

**Keywords:** melanoma, fine needle aspiration, BRAF, immunocytochemistry

## Abstract

Background: Cutaneous malignant melanoma is an aggressive neoplasm. In advanced cases, the therapeutic choice depends on the mutational status of BRAF. Fine needle aspiration cytology (FNA) is often applied to the management of patients affected by melanoma, mainly for the diagnosis of metastases. The evaluation of BRAF mutational status by sequencing technique on cytological samples may be inconvenient, as it is a time and biomaterial-consuming technique. Recently, BRAF immunocytochemistry (ICC) was applied for the evaluation of BRAF V600E mutational status. Although it may be useful mainly in cytological samples, data about BRAF ICC on cytological samples are missing. Methods: We performed BRAF ICC on a series of 50 FNA samples of metastatic melanoma. BRAF molecular analysis was performed on the same cytological samples or on the corresponding histological samples. Molecular analysis was considered the gold standard. Results: BRAF ICC results were adequate in 49 out of 50 (98%) cases, positive in 15 out of 50 (30%) cases and negative in 34 out of 50 (68%) of cases. Overall, BRAF ICC sensitivity, specificity, positive predictive value and negative predictive value results were 88.2%, 100%, 100% and 94.1%, respectively. The diagnostic performance of BRAF ICC results was perfect when molecular evaluation was performed on the same cytological samples. Hyperpigmentation represents the main limitation of the technique. Conclusions: BRAF ICC is a rapid, cost-effective method for detecting BRAF V600E mutation in melanoma metastases, applicable with high diagnostic performance to cytological samples. It could represent the first step to evaluate BRAF mutational status in cytological samples, mainly in poorly cellular cases.

## 1. Introduction

Cutaneous melanoma (CM) is the most aggressive skin neoplasm, responsible for approximately 61,000 deaths per year worldwide [1]. Indeed, CM shows an early tendency to metastasize; thus, lymphatic and hematogenous metastases occur early in tumor progression and the 5-year overall survival (OS) is 23% for advanced melanoma [2]. Beyond cases of clear advanced disease obtained by imaging, the diagnosis of localized or regional recurrence could be very difficult [3]. FNA is a reliable tool to diagnose CM metastases, as it is a rapid and cheap technique [4]. Metastatic CM incidence is less than 5 cases per 100,000 per year [5]. The diagnosis of CM metastases is often based on clinical and instrumental findings and a direct sampling is usually unnecessary. Moreover, deep visceral CM metastases may be difficult to sample by percutaneous techniques such as FNA. So, few studies have evaluated the application of FNA to metastatic CM. However, some reports have shown that the use of FNA with image guidance demonstrates excellent diagnostic performance for the diagnosis of CM metastases, being that the sensitivity (SE) and specificity (SP) range is between 86.5% and 100.0% [3]. A correct sampling of the lesion and adequate cellularity is mandatory to optimize the diagnostic performance of FNC. Particularly, false-negative cases could be related to inadequate sampling. In addition, above all in the cases of a suspicious metastatic site of difficult access, the low cellularity of the samples containing only scattered malignant cells does not permit additional ancillary methods for better characterization [3]. However, when a diagnosis of metastatic CM is posed, the current therapeutic approach is based on advanced medical treatment. Indeed, historically, surgery was the only therapy for CM, and therapeutic options were not available for patients with advanced disease. In the last few years, a better understanding of the molecular landscape of CM led to the development of new therapies for advanced disease. Mitogen-activated protein kinase (MAPK) pathway deregulation has emerged as a main molecular event in the CM oncogenesis, present in up to 70–80% of cases [6]. It is actually known that about 40–50% of CM cases harbor the mutation of the gene for v-Raf murine sarcoma viral oncogene homolog B (BRAF), leading to the BRAF protein constitutive activation, a serine/threonine-protein kinase in the MAPK signaling pathway [7]. The most frequent BRAF mutation is V600E, accounting for approximately 70–80% of cases, followed by V600K, V600D and V600R [7,8]. The identification of BRAF mutations has a predictive value in advanced CM, leading to the selection of patients who may benefit, in the presence of a V600 mutation, from treatment with MAPK inhibitors [9]. Therefore, the molecular workflow for CM should start with the identification of BRAF mutation in stage IIIC or IV CM patients [9]. This evaluation can be carried out using different molecular procedures, including direct sequencing of the PCR product, pyrosequencing, RealTime PCR, molecular hybridization on filter and mass spectrometry [10]. Recently, a BRAF V600E mutation-specific immunohistochemical antibody was introduced, with sensitivity and specificity comparable to the molecular tests in histological samples [11,12,13]. It was consequently proposed, and applied, as a screening tool for a rapid and cheaper assessment of BRAF V600E mutational status in histological samples [11,12,13,14]. In the clinical setting of the advanced disease, the biomaterial obtained from metastasis through FNA procedure for diagnosis could be also used for the assessment of the current BRAF status of the patients, who cannot benefit from surgery with radical intent, avoiding unnecessary surgical stress and public health charges [15,16,17]. Indeed, although the BRAF status usually does not change in the metastasis with respect to the primitive CM, a recent metanalysis underlined the discrepancy between the primitive CM and relative metastasis, calculating a change from BRAF mutation in primitive CM to wild type in relative metastasis in 15.1% of analyzed series and an opposite change in 13.2% of such series [18]. In addition, metastasis could be the only clinical feature of melanoma with an unknown BRAF status. Thus, cytological samples could represent the only biomaterials useful for detecting BRAF status [16,17]. Finally, a different BRAF status of the metastasis with respect to known primitive tumors could also be the expression of another tumor with a different mutational status [19]. For all these reasons, the recommendation of the current guidelines is to determine the mutational status of the metastasis, when possible [20].

However, some limitations about the use of cytological samples for predictive purposes should be addressed. First, the largest limitation to obtain reliable results is the number of neoplastic cells in the cytological sample, sometimes unsatisfactory also for diagnostic purposes [18]. Thus, in such cases, it is required to resort to a surgical sampling of the metastasis for the BRAF status assessment, with further stress for the patient, or to the primitive tumor [21,22]. In addition, in the last case, not only is the BRAF status not always representative of the current status, but when the primitive tumor is a thin melanoma, the number of neoplastic cells may not be enough for the assessment of BRAF mutation with classical extractive methods. Indeed, the immunohistochemical evaluation requires a relatively smaller amount of neoplastic cells, if compared to molecular extractive methods for mutation detections [11,12,13,14]. Thus, in the primitive CM, when very thin or previously highly consumed for diagnosis, BRAF immunohistochemical detection could offer higher diagnostic accuracy than molecular testing [11,12,13,14]. A further limitation is related to the complete consumption of the direct smears of the metastasis cytological sample for DNA extraction, with loss of archival biological material. Finally, molecular analysis is expensive, requires experienced technicians and is not widespread in all laboratories worldwide. Thus, although the application of molecular analysis to cytological samples is generally successful, the use of immunocytochemistry (ICC) for predictive tests in advanced CM on cytological samples could present some advantages.

The aim of our study is the evaluation of the diagnostic performance of BRAF ICC on a retrospective series of metastatic CM diagnosed by FNA, comparing ICC results with the gold standard molecular analysis results.

## 2. Materials and Methods

### 2.1. Patients Selection

All cases of US-guided or CT-guided FNA in patients with previous CM and clinical suspicious metastasis in the follow-up between January 2017 and December 2020 from the archives of the Pathology Unit of University “Vanvitelli” Hospital in Naples (Italy) were revised. Negative cases were excluded. Positive cases were selected in our series according to the following criteria: (1) diagnosis of CM metastases rendered on FNA samples; (2) the realization of a cell-block (CB); (3) the presence of residual biomaterial in the CB; (4) molecular evaluation of BRAF mutational status performed on the same cytological sample or the corresponding histological sample, when surgery was performed.

Fifty-seven positive consecutive cases were initially retrieved. All the cases were reviewed by two expert cytopathologists (MM, IC) to confirm the diagnosis and to assess the presence of sufficient residual tumor cells in the CB. The diagnosis was confirmed in all cases. Two cases were excluded because they did not include a CB, and 4 cases were excluded because the CB did not include residual biomaterial. One case was excluded because a molecular evaluation was not possible, as this case only included few neoplastic cells. Therefore, the final number of cases included in the series was 50. Each case included direct smears-stained by Diff Quick and Papanicolaou and a CB. Clinical and molecular data were collected from the archives of the Pathology Unit.

### 2.2. BRAF V600E Immunocytochemistry Technique

ICC was performed on 4 μm-thick FFPE cell-block slices using a fully automatized assay based on the Ventana^®^ BRAF V600E (VE1, Ventana-Roche Diagnostics, Meylan, France) mouse monoclonal primary antibody in combination with the Ventana OptiView DAB IHC Detection Kit^®^ on the Ventana^®^ Benchmark XT platform (Ventana-Roche Diagnostics, Meylan, France). The procedure was performed according to the manufacturer’s instructions.

### 2.3. BRAF Immunocytochemistry Evaluation

All immunostained slides were evaluated by two cytopathologists in absence of any information about molecular data. Immunostaining was primarily interpreted as positive or negative. We defined a case as positive if it showed diffuse cytoplasmic staining, according to data reported in histological series [13,23]. We considered a case as negative if no staining or only nuclear dot staining was present. Furthermore, the percentage of positive neoplastic cells and intensity of the staining were recorded. The percentage of positive neoplastic cells were calculated by comparing the stained neoplastic cells to the total number of neoplastic cells in the slide.

### 2.4. BRAF Molecular Analysis

According to strategies followed during clinical management, molecular analyses were previously performed preferentially on cytological samples; when the number of neoplastic cells was not enough for extractive-based molecular testing, the analysis was conducted on the histological sample of the CM metastasis when the patients were submitted to surgery or on the primitive tumors if tissue from metastases were not available.

The evaluation of the mutational status of the BRAF gene was performed by the NGS method. DNA was extracted from 4 unstained 10 µM FFPE tissue sections or from the cytological samples. DNA was obtained using the QIAamp^®^ DNA FFPE kit Tissue (Qiagen, Hilden, Germany) for histological samples or using the Qiagen QIAamp^®^ DNA Micro kit. (Qiagen, Hilden, Germany) for cytological samples, according to the manufacturer’s instructions. The massive parallel sequencing of DNA libraries by ION Torrent Personal was used as previously reported [7]. Sequencing was carried out using different chips on the Ion Personal Genome Machine System (PGM™, Thermo Fisher Scientific, Waltham, MA, USA). Torrent Suite Software v.4.0.2 (Life Technologies, Carlsbad, CA, USA) to assess run performance and data analysis was used. Integrative Genomics Viewer (IGV v 2.2, Broad Institute, Cambridge, MA, USA) was used for visual inspection of the aligned reads. Data were analyzed using Ion Reporter software [22] and further filtered through quality checking. We selected all SNVs in the studied genes resulting in a non-synonymous amino acid change, or a premature stop codon, and all short indels resulting in either a frameshift or insertion/deletion of amino acids. All SNVs were analyzed for previously reported hotspot mutations (somatic mutations reported in COSMIC database) and novel variations, i.e., new mutations detected by NGS but not reported in either COSMIC or db SNP databases.

### 2.5. Evaluation of Diagnostic Performance of BRAF Immunocytochemistry

BRAF ICC SE and SP were calculated considering the molecular evaluation of BRAF mutational status as the gold standard.

### 2.6. Ethical Consideration

The present study was retrospectively conducted on archival biological samples. Both the cytological and histological diagnoses, as well as molecular analysis of BRAF mutational status, had already been rendered in all cases. At the time of the FNA procedure, a written consent, including the consent to use the diagnostic data for scientific purposes, had been obtained from each patient. The approval by the institutional ethical board was collected.

## 3. Results

### 3.1. Clinic-Pathological Data

This series included 50 cases diagnosed as CM metastases on cytological samples obtained by FNA. The series included 38 (76%) males and 12 (24%) females (M:F = 3.2:1), with a mean age of 62 years (range from 38 to 86 years). The CM metastases were located at lymph nodes (LNs) in 43 out of 50 (86%) of cases, at subcutaneous nodules in 4 out of 50 (8%) of cases and at lungs in the remaining 3 (6%) of cases. In detail, the location of CM metastases was: right axillary LNs in 14 cases, left axillary LNs in 8 cases, right inguinal LNs in 8 cases, left inguinal LNs in 4 cases, right cervical LNs in 5 cases, left intraparotid LNs in 2 cases, right intraparotid LN in 1 case, deep para-aortic LN in 1 case, right lung in 2 cases, left lung in 1 case, subcutis in 4 cases. Clinical findings are summarized in Table 1.

### 3.2. Braf V600E ICC and Molecular Detection

BRAF ICC was performed on CB sections in all cases. Molecular evaluation of the BRAF status was performed in all cases, particularly on cytological samples in 17 cases and on histological samples in 33 cases (primitive or metastasis). BRAF ICC resulted adequate in 49/50 (98%) cases, positive in 15 out of 50 (30%) cases and negative in 34 out of 50 (68%) of cases (Figure 1).

The percentage of positive neoplastic cells ranged from 40% to 100%, with the mean positivity of 75.3%, and the median positivity of 80% (Figure 2). The staining intensity resulted heterogeneous, ranging from slight to strong intensity (Figure 3).

BRAF V600E mutation was confirmed by molecular analysis in all the 15 BRAF ICC-positive cases. The analysis was performed on the corresponding histological samples (2 cases on the primitive neoplasms and 8 cases on the metastatic surgical biopsies) in 10 cases, and on the cytological samples in 5 cases. One out of 15 BRAF ICC-positive cases showed both V600E and V600D mutations at the molecular analysis.

Concerning the 34 negative cases, molecular analysis was performed on corresponding histological samples in 22 cases (14 cases on the primitive neoplasms and 8 cases on the metastatic surgical biopsies) and on cytological samples in 12 cases. The molecular analysis confirmed the absence of BRAF V600E mutation in 32 out of 34 (94.1%) cases. BRAF V600E mutation was instead detected by molecular analysis in 2 out of 34 (5.9%) ICC-negative cases.

In one case (2%), the BRAF ICC test was considered inadequate because of the presence of abundant melanin pigment that invalidated the evaluation.

BRAF molecular analysis resulted in inadequate in one case, in which the analysis was performed on a histological sample of the primitive neoplasm. In this case, the cellularity of the cytological sample was not sufficient for the molecular test, and the patient was not submitted to surgery for the removal of the metastases. Moreover, the biological sample of the primitive neoplasm resulted inadequate for the molecular test, as it was a thin CM and the biomaterial was largely consumed for diagnostic tests (multiple levels, immunohistochemical tests).

BRAF mutations different from V600E were detected by molecular analysis in two cases and included V600K in one case and V600R in the other case.

### 3.3. Diagnostic Performance of BRAF ICC

Considering BRAF V600E mutation, overall BRAF ICC showed a SE of 88.2% and an SP of 100%. The positive predictive value (PPV) was 100% and the negative predictive value (NPV) was 94.1%. We can define two different sub-groups in the series: (1) BRAF V600E ICC vs. molecular evaluation on histological samples and (2) BRAF V600E ICC vs. molecular evaluation on cytological samples. Indeed, molecular analysis was performed on histological samples when the cytological samples were not sufficient for both diagnostic and predictive purposes.

Diagnostic performance of BRAF ICC resulted differently in these two sub-groups, being higher in the subgroup of cytological samples. Indeed, in the first group, the SE, SP, PPV, and NPV were 83.3%, 100%, 100%, and 90.9%, respectively. While, in the second group, the SE, SP, PPV and NPV were 100%, 100%, 100% and 100%, respectively. Considering all the BRAF mutations, and including thus also BRAF mutations other than V600E, SE, SP, PPV and NPV were 78.9%, 100%, 100% and 88.2%, respectively.

Results about the diagnostic performance of BRAF ICC are summarized in Table 2.

Some examples of BRAF ICC staining are shown in Figure 4.

## 4. Discussion

Therapy of advanced CM has dramatically changed in the last few years. Nowadays, MAPK inhibitors and immunotherapy are applied worldwide, significantly improving the overall survival of the patients [24]. Consequently, the evaluation of BRAF mutational status has become a milestone in the management of patients with advanced CM, directly affecting the therapy choice. Indeed, BRAF inhibitors in combination with MEK inhibitors represent the standard treatment for patients with advanced CM carrying BRAF mutations [25]. Overall, BRAF mutations occur in about 50% of all CMs but are more frequent in CM developing in intermittently sun-exposed skin [24]. The most common BRAF mutation in CM, accounting for about 70–80% of the mutated cases, is BRAF V600E, a single nucleotide mutation resulting in the substitution of valine with glutamic acid. The V600E-mutated BRAF constitutively activates MEK regardless of RAS signaling. Other less common BRAF mutations may occur in CM, including V600K (accounting for about 5–6% of cases), with valine replaced by lysine, V600R and V600D [26].

As the mutational status of the neoplasms may be discordant between the primary neoplasms and the metastases in the same patient [18], the actual guidelines recommend defining the mutational status of the metastases, when it is possible [20]. However, the management of the biomaterial is a hot issue in advanced melanoma, and it is often the main limitation for the predictive tests. Indeed, the direct sampling of the metastases is usually unnecessary for the diagnosis, which is usually based on clinical and instrumental findings. On the other hand, the execution of invasive surgery only to obtain biomaterial for predictive purposes may be problematic and not convenient for patients with low-performance status and deep metastases. In this setting, FNA represents a useful tool in the management of patients with advanced CM, for both diagnostic and predictive purposes. Indeed, FNA may be used to confirm the diagnosis of CM metastases in patients with a known history of CM or to obtain the diagnosis in patients with metastases of unknown primary neoplasm, with high sensitivity and specificity [4]. In addition, FNA may be used to collect biomaterial for the predictive evaluation of BRAF mutational status in patients with a clinically known CM metastasis, as recommended also by recent National Comprehensive Cancer Network guidelines [9]. Interestingly, our series included three cases located in the lungs, three cases located in the intra-parotid lymph nodes and one case located in a deep para-aortic lymph node. In all these cases, sampling biomaterial by FNA rather than by surgery certainly resulted in the best options for the patients. However, the amount of available biomaterial in cytological samples may be a limitation for predictive purposes. Indeed, the diagnosis of CM is challenging, and a variable but significant percentage of the biomaterial could be used for the diagnosis. In these circumstances, the cytological sample may not be sufficient for DNA-based molecular techniques for predictive purposes, which are consequently performed on the primary neoplasm, even if the molecular status could be different between primary neoplasm and metastases. In this clinical scenario, ICC could play an important role to obtain information about BRAF V600E mutation in CM metastases using cytological samples not adequate for DNA-based molecular techniques. Although DNA-based molecular techniques are the gold standard to define BRAF mutational status, BRAF ICC was recently introduced as a rapid test to evaluate the presence of BRAF V600E mutation, demonstrating high diagnostic performance in histological samples [11,12,13,14]. However, although it is well-known that FNA samples may be used for molecular analysis of BRAF mutations by DNA-based methods, data about the performance of BRAF ICC on CM cytological samples are missing.

We tested BRAF ICC on a series of 50 FNA samples of CM metastases, comparing the ICC results with the gold standard molecular analysis. In this series, we observed 15 out of 50 (30%) ICC-positive cases. All the ICC-positive cases were confirmed by the molecular analysis, and the series did not include false-positive cases, resulting in an SP and a PPV of 100%. Of the 34 ICC-negative cases, 2 cases showed the BRAF V600E mutation by molecular analysis, representing 2 false-negative cases. False negativity could be due to defects of the technique, or to defects of sampling. Overall, BRAF ICC demonstrated a high diagnostic performance, with an SE of 88.2%, a SP of 100%, a PPV of 100% and a NPV of 94.1%. Our results are comparable to the data reported in the literature regarding BRAF ICC applied to histological samples. Indeed, in the studies about diagnostic performance of BRAF ICC performed on histological samples, the SE ranges from 85% to 100% and the SP ranges from 93% to 100% [11,12,14,23,27,28,29,30,31,32,33,34,35].

In our series, BRAF ICC interpretation resulted relatively easily in most cases, as the intensity of the staining was high or moderate in 13 out of 15 (86.7%) positive cases. Moreover, a high number of neoplastic cells (often, most of the neoplastic cells) resulted stained in the positive cases, as the least percentage of positive cells was 40%, the mean percentage of positive cells was 75.3% and the median percentage of positive cells was 80%. However, attention must be paid to the evaluation of the overly-pigmented cases. Indeed, BRAF ICC shows cytoplasmic staining and consequently, the staining may overlap with melanin pigment. Our series included four (8%) cases characterized by intense and diffuse pigmentation. For these cases, a collegial evaluation of the BRAF immunostained section and the corresponding H and E-stained section of the corresponding CB was performed. In three of the four cases, the evaluation of the immunocytochemical signal was possible thanks to the presence on the immunostained sections of clearly negative malignant cell groups (Figure 4).

For these cases, the molecular evaluation confirmed the negativity of the test. In the fourth case, the abundant melanic pigment in the cytoplasm and in the background, together with a discohesive pattern, did not allow a clear evaluation of the immunocytochemical signal and therefore this case was considered inadequate for ICC evaluation (Figure 5). Molecular analysis should always be considered in heavily pigmented cases, to avoid the risk of false-positive results.

Moreover, it should be kept in mind that BRAF ICC only stains cases harboring V600E mutation, but it does not stain cases harboring other BRAF mutations. In our series, indeed, BRAF mutation other than V600E results were present in two cases (one case with V600K mutation and one case with both V600K and V600R mutations). Considering the possibility of BRAF ICC false-negative cases, and the possibility of BRAF mutations other than V600E in ICC negative cases, BRAF ICC should be applied to FNA samples as a screening tool, and molecular analysis should be considered mandatory in BRAF ICC negative cases. We propose an algorithm in which BRAF ICC represents the first step for the evaluation of BRAF mutational status in cytological samples. A positive result could be adequate to address the patient to target therapy, while molecular analysis should be performed in BRAF ICC negative cases (Figure 5).

This algorithm may be particularly useful in poorly cellular samples. Indeed, the cytological samples sometimes may be poorly cellular, and consequently not suitable for extraction-based molecular analysis, forcing the clinician to obtain a surgical excision of the metastasis, or the pathologists to perform the predictive tests on the primitive neoplasms. Moreover, even in the primitive melanoma, the neoplastic component could be scant, making the identification of BRAF mutation unsuccessful. In these settings, the chance offered by ICC on cytological samples from CM metastases could be a valid opportunity for the therapeutic strategy of the patients.

## 5. Conclusions

In conclusion, BRAF ICC is a rapid, cost-effective method for detecting BRAF V600E mutation in MM metastases, applicable with high diagnostic performance to cytological samples. It could represent the first step to evaluate BRAF mutational status in cytological samples, mainly in poorly cellular cases.

## Figures and Tables

**Figure 1 diagnostics-11-01110-f001:**
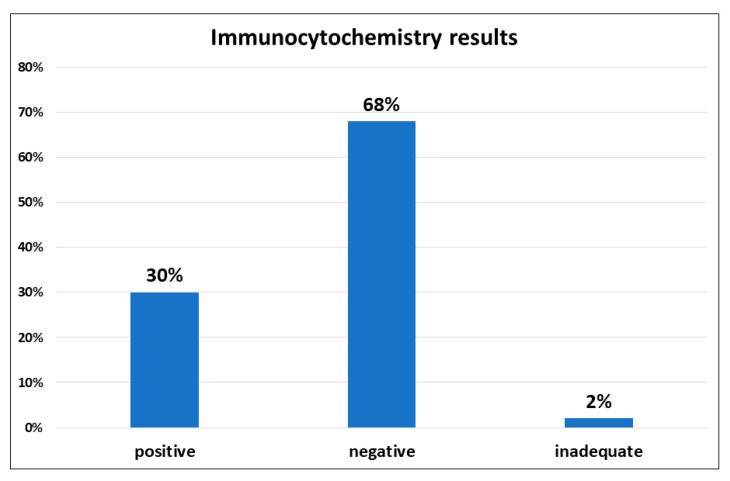
BRAF V600E mutation-specific adequacy. The immunocytochemistry resulted in adequate in 98%, and only 1 out of 50 (2%) cases resulted in inadequate. BRAF Immunocytochemistry resulted positive in 30% of the cases and negative in the remaining 68%.

**Figure 2 diagnostics-11-01110-f002:**
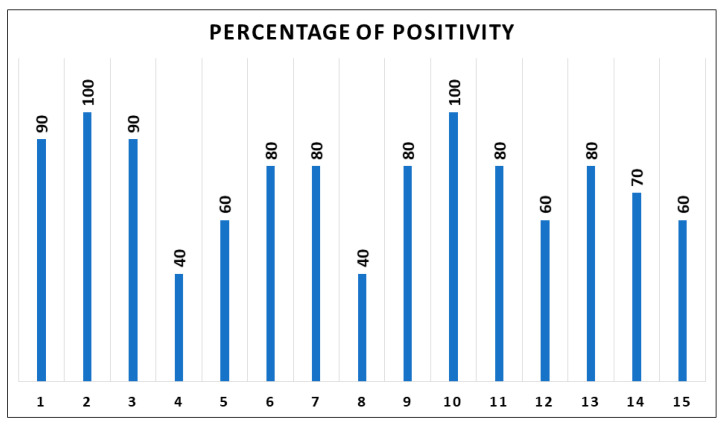
Percentage of BRAF immunocytochemistry-positive cells in the 15 positive cases.

**Figure 3 diagnostics-11-01110-f003:**
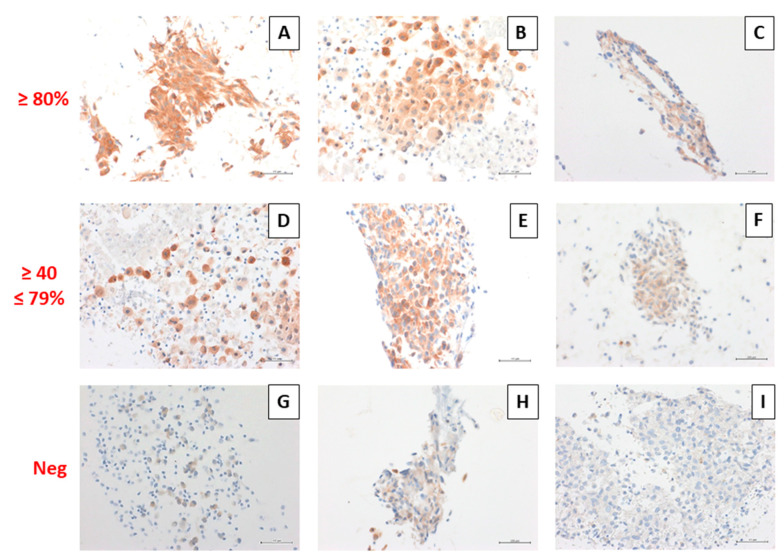
Overview of BRAF V600E immunocytochemistry on CB sections. Granular cytoplasmic positivity with different intensity of the immunocytochemical signal: strong intensity (**A**,**D**), moderate intensity (**B**,**E**), slight intensity (**C**,**F**) in over 80% of the neoplastic cells (**A**–**C**) and between 40 and 79% of the neoplastic cells (**D**–**F**) Negative cases (**G**–**I**). (Immunocytochemical stains, original magnification 400×).

**Figure 4 diagnostics-11-01110-f004:**
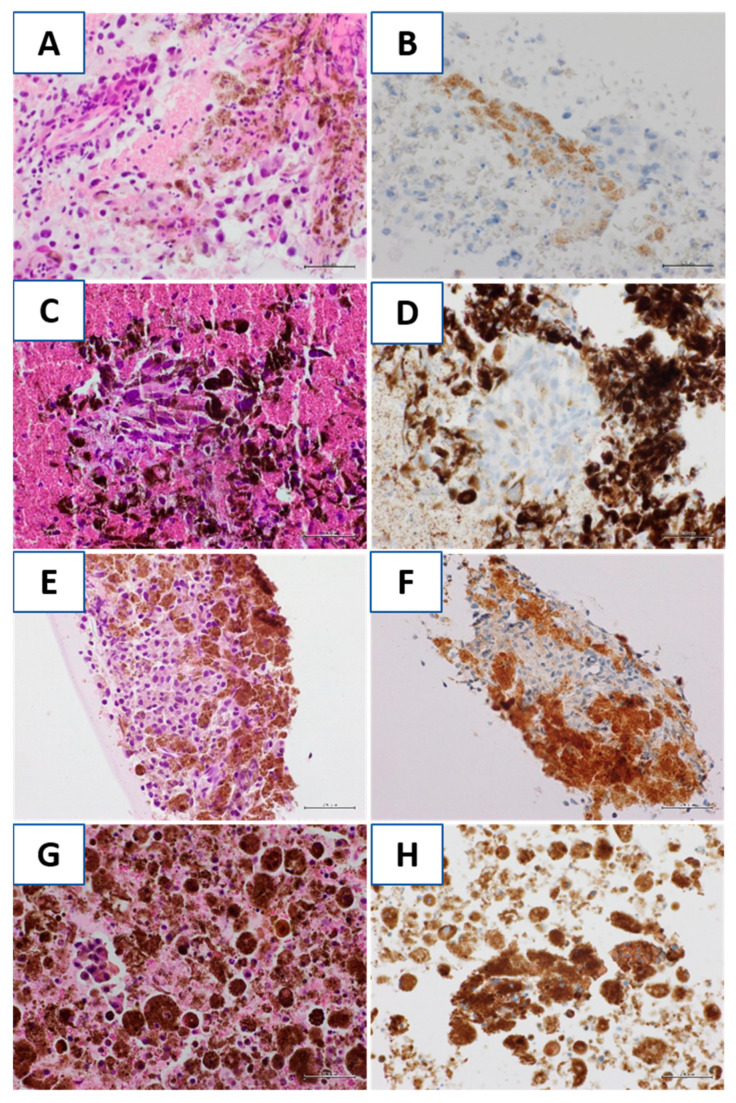
(**A**,**B**) Case 1, presence of melanic pigment in the cytoplasm of histiocytes and melanoma cells on cell-block section. BRAF ICC was negative in malignant cells. In this case, at the molecular assessment, BRAF resulted from wild-type; (**C**,**D**) Case 2, a melanoma cellular group in a proteinaceous background with melanic pigment on cell-block section. BRAF ICC was negative in the malignant group. At the molecular assessment, BRAF was wild-type; (**E**,**F**) Case 3, a cellular aggregate of melanoma cells in a melanic background on cell-block section. BRAF ICC was negative in the neoplastic cells. At the molecular assessment, BRAF V600R mutation was detected; (**G**,**H**) Case 4, numerous engulfing histiocytes melanic pigment, which is also present in the background and few melanoma cells were present on cell-block section. BRAF ICC was considered not evaluable. At the molecular assessment, a BRAFV600E was detected. (Hematoxylin-eosin and immunocytochemical stains, original magnification 400×).

**Figure 5 diagnostics-11-01110-f005:**
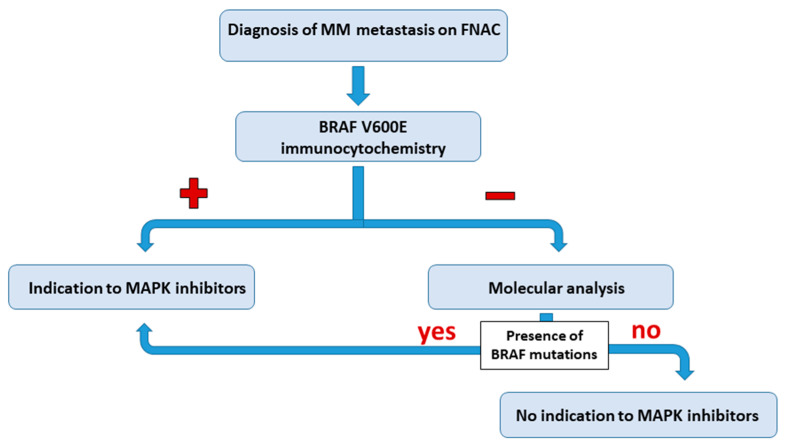
Proposed algorithm for the evaluation of the BRAF V600E mutation on a cytological sample.

**Table 1 diagnostics-11-01110-t001:** Clinical findings.

Total Cases	50
Sex	
Males	38(76%)
Females	12 (24%)
Males:Females	3.2:1
Age	
Range	38–86
Mean age	62
Location	
Right axillary lymph node	14 (28%)
Left axillary lymph node	8 (16%)
Right inguinal lymph node	8 (16%)
Left inguinal lymph node	4 (8%)
Right cervical lymph node	5 (10%)
Intra-parotid lymph node	3 (6%)
Para-aortic lymph node	1 (2%)
Lung	3 (6%)
Subcutis	4 (8%)

**Table 2 diagnostics-11-01110-t002:** Diagnostic performance of BRAF V600E mutation-specific immunocytochemistry.

	Overall Series	Molecular Tests on Histology	Molecular Tests on Cytology
Tested cases	50	33	17
Adequacy	49/50 (98%)	32/33 (96.9%)	17/17 (100%)
True-Positive	15	10	5
False-Positive	0	0	0
True-Negative	32	20	12
False-Negative	2	2	0
Sensitivity	88.2%	83.3%	100%
Specificity	100%	100%	100%
PPV	100%	100%	100%
NPV	94.1%	90.9%	100%
Accuracy	95.1%	93.7%	100%

Abbreviations: PPV: positive predictive value; NPV: negative predictive value.

## Data Availability

The data presented in this study are available on request from the corresponding author.

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
