# Peer review of "Predictive Evaluation on Cytological Sample of Metastatic Melanoma: The Role of BRAF Immunocytochemistry in the Molecular Era"

_diagnostics, 2021, doi:10.3390/diagnostics11061110_

Round 1

Reviewer 1 Report

The paper addresses in a well-structured manner to the problematic of evaluation of metastatic melanoma from a molecular point of view.

The authors have to be respond at some minor issues:

  • Line 103: are the “two expert cytopathologists” among authors? If YES, please specify clearly that in the Author contributions section (lines 318-323). If NO, please add an Acknowledgment section to thanks them!
  • Lines 144-145: please elaborate on “quality checking”
  • Please ad an empty row before Table 2
  • Figure 6: there is a small square below “Diagnosis of MM metastasis on FNAC” –what is its purpose?

Reviewer 2 Report

The authors examine the feasibility of using a BRAF V600E-specific monoclonal antibody to identify melanoma patients who could benefit from targeted therapies, such as MAPK inhibitors. The number of cases is particularly small considering the prevalence of metastatic melanoma, and this impacts the statistical accuracy noticeably. It is unclear from their results what benefit such a test would offer patients.

  1. One of the histology samples was inadequate for molecular testing. The authors should explain whether the problem was in the sample or the extraction technique.
  2. The authors must include the two cases with V600K and V600R mutations in their false negative calculations: these mutations respond to MAPK inhibitors, which is the point of the test.
  3. The number of cases is too small to do accurate statistics comparing the different tests, especially when the molecular testing was performed on two different types of samples. The only clinically significant result is that ICC failed to identify four cases that were positively identified by molecular testing.
  4. It is unclear why the authors graded the positive staining: all grades of staining were considered positive, and staining grade was not correlated with molecular success or any other diagnostic variables.
  5. The Flowchart (figure 6) shows that over 2/3 of cases (ICC negative) would still require molecular testing; so, why not simply do molecular testing on everyone? ICC seems like an unnecessary extra step.
  6. The best “fit” for using BRAF(V600E)-specific ICC testing would be in underserved communities as an inexpensive and technically simple way to identify patients with high-risk metastatic melanoma. But, many false-negative cases would still fall through the cracks, and many positive patients would not have access to gene-targeted therapies like MAPK inhibitors. Further, the authors have not established a specific, uniform protocol that could be implemented for scoring by other centers. From a diagnostic perspective, using the antibody for an ELISA-based test would vastly improve throughput and reproducibility and eliminate subjectivity.
